# Differential Laboratory Diagnosis of Acute Fever in Guinea: Preparedness for the Threat of Hemorrhagic Fevers

**DOI:** 10.3390/ijerph18116022

**Published:** 2021-06-03

**Authors:** Vladimir G. Dedkov, N’Faly Magassouba, Olga A. Stukolova, Victoria A. Savina, Jakob Camara, Barrè Soropogui, Marina V. Safonova, Pavel Semizhon, Alexander E. Platonov

**Affiliations:** 1Pasteur Institute, Federal Service on Consumers’ Rights Protection and Human Well-Being Surveillance, 197101 Saint Petersburg, Russia; viksavina@yandex.ru (V.A.S.); ov_plt@mail.ru (A.E.P.); 2Martsinovsky Institute of Medical Parasitology, Tropical and Vector Borne Diseases, Sechenov First Moscow State Medical University, 119435 Moscow, Russia; 3Laboratoire de Virologie, Université Gamal Abdel Nasser de Conakry, Projet de Recherche sur les Fièvres Hémorragiques en Guinée, 001 B1568 Conakry, Guinea; magasoubanfaly@gmail.com (N.M.); jacob2240@gmail.com (J.C.); vgdedkov@mail.ru (B.S.); 4Central Research Institute for Epidemiology, Federal Service on Consumers’ Rights Protection and Human Well-Being Surveillance, 111123 Moscow, Russia; ovasika@yandex.ru; 5Anti-Plague Center, Federal Service on Consumers’ Rights Protection and Human Well-Being Surveillance, 119121 Moscow, Russia; marina-iris@mail.ru; 6The Republican Research and Practical Center for Epidemiology and Microbiology, 220114 Minsk, Belarus; pavel5555@tut.by

**Keywords:** Guinea, zoonotic pathogens, acute febrile illness, diagnostics, fever

## Abstract

Acute febrile illnesses occur frequently in Guinea. Acute fever itself is not a unique, hallmark indication (pathognomonic sign) of any one illness or disease. In the infectious disease context, fever’s underlying cause can be a wide range of viral or bacterial pathogens, including the Ebola virus. In this study, molecular and serological methods were used to analyze samples from patients hospitalized with acute febrile illness in various regions of Guinea. This analysis was undertaken with the goal of accomplishing differential diagnosis (determination of causative pathogen) in such cases. As a result, a number of pathogens, both viral and bacterial, were identified in Guinea as causative agents behind acute febrile illness. In approximately 60% of the studied samples, however, a definitive determination could not be made.

## 1. Introduction

Reliable and detailed epidemiological data are essential for both low-income and high-income countries. The best way to get this kind of data is likely through international collaboration. This paper presents the results of a cooperative study involving specialists from Guinean University and five leading Russian and Belarusian epidemiological research institutions.

The Republic of Guinea is a West African country with an area of 245,857 km^2^. According to the World Health Organization (WHO), the country’s population (2016) is approximately 12.4 million people [1]. Guinea is one of the poorest countries in the African region and, as a result, a low level of public health continues to be a problem, as reflected by several indicators. According to the WHO, the average life expectancy in Guinea (2016) is 59.8 years [2]. The infant mortality rate (for children aged 5 and below) is 91.7 per 1000 thousand live births [3].

The top causes of illness and death in children include malaria, respiratory illness, intestinal diseases, tuberculosis, HIV infection, measles, parasitic illnesses, and infectious diseases of unknown etiology. Among the adult Guinean population, the greatest health burdens are HIV/AIDS, malaria and tuberculosis (a combined incidence of approximately 1500 per 100,000 population), acute respiratory illnesses (approximately 1000 per 100,000), and other infectious diseases (approximately 1400 per 100,000) [1,4]. It should also be noted that the above data are likely underestimates. In most cases, it is not possible to establish a precise diagnosis due to a lack of qualified medical personnel and necessary diagnostic laboratory facilities, particularly in remote areas of the country.

In addition, the situation in Guinea is complicated due to numerous, widespread zoonotic pathogens (both viral and bacterial), including Yellow Fever Virus (YFV); Dengue Virus (DENV); West Nile Virus (WNV); Chikungunya Virus (CHIKV); Crimean-Congo Hemorrhagic Fever Virus (CCHFV); Rift Valley Fever Virus (RVFV); Lassa Virus (LASV); spotted fever group rickettsias (SFG rickettsias, including *Rickettsia africae*, *R. aeschlimannii*, *R. massiliae*); relapsing fever *Borrelia* spp.; *Bartonella* spp.; *Anaplasmataceae* bacteria; *Coxiella burnetii;* and poorly studied viruses such as O’nyong-nyong virus (ONNV) and Tahyna virus (TAHV), etc. These cause diseases of varying severity with similar clinical symptoms [5,6,7,8,9].

Timely detection and diagnosis of the above diseases are important in themselves, but it is also important that they are able to masquerade the early manifestations of filovirus fever epidemics, which are characterized by high mortality and rapid spreading. For example, the largest recorded Ebola outbreak in history (between 2014 and 2016), accounting for 11,323 deaths among 28,646 confirmed cases, can be cited [10]. Notably, the first cases associated with that epidemic are now known to have occurred in 2013 [11]. From an epidemiological point of view, infection patterns have changed, and the vast majority of cases occurred through person-to-person contact [12]. The disease spread widely, including urban population involvement, due to a lack of diagnostic tools, similarities between Ebola virus disease (EVD) and other diseases which could clinically present with hemorrhagic syndromes and/or fever, as well as a lack of vigilance by health authorities [13].

Moreover, the gaps in our understanding of epidemic threats in Africa is not solely an African problem. Numerous cases of importation of African zoonotic infections to Europe or the Americas have been well documented. In the worst case, they may lead to autochthonous outbreaks or even permanent establishment of an infection in a new area [14,15,16]. As cases of imported infection are rare and their initial symptoms are often unspecific, such diseases can be easily missed during a patient’s differential diagnostics.

It would be inaccurate to say that infectious disease incidence in Guinea has not been studied at all. Indeed, the medical literature does reflect numerous publications devoted to various aspects of infectious disease in the region. These works, however, have been narrow in focus. They have mainly centered on the study of infectious agent biological properties, organization and conduct of specific prevention, or the clinical course and treatment of West African diseases. We utilized molecular and serological methods to analyze numerous samples from acute fever patients with the goals of determining which infections are most relevant to Guinea and which require improved differential diagnosis relative to hemorrhagic fevers, including EVD.

## 2. Materials and Methods

### 2.1. Sampling

Serum samples were collected from December 2016 to April 2017 from patients unvaccinated against YFV displaying symptoms of an acute febrile illness. The patients were hospitalized in different regions of the country, and their samples were sent for study to the Virology Laboratory of Hemorrhagic Fevers Research Project, located in Gamal Abdel Nasser University (Conakry, Guinea) in 2016–2017.

The samples were collected in 25 prefectures representing lower, middle, upper, and forested Guinea (see Figure 1). Serum samples were collected on the first or second day of hospitalization, and malaria was excluded at the point of care using a rapid diagnostic test. Onset of illness, however, was between 4 and 11 days prior to sampling. Venous blood samples (5 mL) were collected in VACUETTE Serum Fast Separator tubes (Greiner Bio One, Kremsmünster, Austria), kept at room temperature for 10 min, and centrifuged at 3000× *g* for 5 min. Serum samples were then placed in 1.5 mL tubes and delivered to the laboratory within two days after sampling; portable containers featuring frozen-block cooling were used (temperature not higher than 8 °C). Delivered samples were stored at −70 °C until analysis (storage from one month up to one year). Repeated freeze–thaw cycles were avoided during the transportation and testing process. To ensure patient confidentiality, each clinical record was assigned an individual number, which was subsequently used to label tubes containing serum for serological study.

### 2.2. Nucleic Acid Extraction

Viral RNA was extracted from 140 µL of undiluted sera using a QIAamp viral RNA kit (Qiagen, Hilden, Germany) according to the manufacturer’s instructions.

### 2.3. Molecular Analysis Using Real-Time Polymerase Chain Reaction

Molecular methods were used to study all of the samples in terms of Ebola virus disease (EVD), Marburg fever disease (MFD), Crimean-Congo hemorrhagic fever disease (CCHFD), Lassa fever disease (LFD), Denge fever disease (DFD), and West Nile fever disease (WNFD) (Table 1).

All samples were studied for EVD, MFD, CCHFD, DFD, and WNFD using the following commercial kits: CCHFV-FL Kit (Amplisens^®^, Moscow, Russia); Dengue virus-FL (Amplisens^®^, Moscow, Russia); WNV-Fl Kit (Amplisens^®^, Moscow, Russia); and *Filo*A-screen-FL and *Filo*B-screen-FL Kits (Amplisens^®^, Moscow, Russia) for the diagnosis of all pathogenic filoviruses and non-pathogenic Reston Ebolavirus [17]. All analyses were performed according to the manufacturers’ instructions. Analysis for LFD was conducted using Qiagen One-Step RT-PCR Kit reagents (Qiagen, Germany) as described by *Ölschläger* et al. [18] with minor modifications. In brief, GPC gene fragments (300 bp) were amplified in a 25 μL reaction containing 5 μL of viral RNA, 0.6 μM of S36 [19] and 0.6 μM of LVS-339-rev (GTTCTTTGTGCAGGAMAGGGGCATKGTCAT) [20] primers, 0.4 mM dNTPs, 5 μL of 5x OneStep buffer, and 1.0 μL of enzyme mix. Thermal cycling parameters were as follows: 50 °C for 30 min, 95 °C for 15 min, followed by 45 cycles of amplification (95 °C for 30 s, 52 °C for 30 s, and 72 °C for 30 s). A final elongation was performed at 72 °C for 5 min. Following electrophoretic separation on a 2% agarose gel, amplified PCR products were visualized with ethidium bromide using UV illumination.

### 2.4. Serological Study Using Enzyme-Linked Immunosorbent Assay

ELISA was used for yellow fever diagnosis and Lassa fever confirmation. Detection of serum anti-*YFV* IgM was performed using the MAC-ELISA IgM capture assay, developed by the Center for Disease Control and Prevention (CDC, Atlanta, GA, USA), as recommended for WHO Yellow Fever network laboratories [20]. Specifically, wells of MaxiSorp 96-well flat-bottomed strips (Nunc, Roskilde, Denmark) were coated overnight at 4 °C with 75 μL/well of goat anti-human IgM (Sigma-Aldrich, St. Louis, MO, USA) diluted at 1:2000 in carbonate buffer (pH 9.6), followed by 4 washes with Wash Buffer (PBS with 0.05% Tween 20) and a 1 h blocking step using 200 μL/well of Blocking Buffer (10% horse serum in PBS/0.05% Tween 20/1% nonfat milk). After 4 plate washes with Wash Buffer, 75 μL/well of patient serum, the positive control (CDC, USA), or the negative control (CDC, USA) was added to wells and incubated for 1 h at 37 °C; all samples and controls were diluted at 1:400 in Dilution Buffer (PBS/0.05% Tween 20/1% nonfat milk).

Each diluted sample was added to two separate wells for future incubation with the yellow fever antigen and control antigen. In parallel, the yellow fever antigen or control antigen (CDC, USA) were reconstituted in Dilution Buffer (1:40). After 4 plate washes, the yellow fever antigen and control antigen were added to their separate partner wells (75 μL/well), followed by a 1 h incubation at 37 °C. After 4 plate washes, the pan-flavivirus 6B6C-1 HRP-conjugated mAb (CDC, USA), pre-diluted at 1:6000 in Dilution Buffer, was added and incubated for 1 h at 37 °C. After 5 plate washes, 10 μL/well of Enhanced K-Blue TMB substrate (Neogen Corp., Lexington, KY, USA) was added. Plates were incubated for 10 min at room temperature, after which the reaction was stopped by addition (50 μL/well) of 1 M H_2_SO_4_. Optical absorbance was measured at 450 nm. Detection of serum anti-*LASV* IgM was performed using the ReLASV^®^ Pan-Lassa IgG/IgM ELISA Test Kit based on a GP-linked protein (Zalgen Labs LLC, Germantown, MD, USA), according to the manufacturer’s instructions.

### 2.5. Serological Study Using Protein Microarray

#### 2.5.1. Design of the Planar Protein Microarray

The presence of specific IgM antibodies in patient sera (anti-*DENV*, anti-*ZIKV*, anti-*WNFV*, anti-*CCHFV*, anti-*CHIKV,* anti-*RVFV*, anti-Z*EBO*V (Zaire Ebola virus), anti-*MARV* (Marburg virus), anti-*SFG rickettsia*, and anti-*Borrelia* spp.) was determined using a microarray containing the following recombinant antigens: DENV type 1 E protein; DENV type 1 NS1 protein; DENV type 3 E protein; DENV type 3 NS1 protein; ZIKV E protein; ZIKV NS1 protein; CHIKV E1-protein (Meridian Life Science, Memphis, TN, USA); ZEBOV NP; MARV NP (The Research and Practical Center for Epidemiology and Microbiology, Minsk, Belorussia) [21]; CCHFV fragments of glycoprotein G1 (AA 1451–1469, 1451–1469, and 1613–1631) and fragment of L protein (AA 859–873); for RVFV, the NP protein, its NPsh fragment (AA 121–201), and a G2 glycoprotein fragment (AA 522–535); CHIKV E2 protein fragment (AA 1–264); WNFV NS1 protein (all from the Central Research Institute for Epidemiology (CRIE), Moscow, Russia) [22]; *B. afzelii* and *B. garinii* proteins p100, p41, p39, p58, BBK32, OspC, p17, and the antigenic fragment of the VlsE protein; *B. miyamotoi* GlpQ, Vsp1, Vlp15/16, Vlp18, and Vlp5 proteins (all from CRIE, Moscow, Russia); and *R. sibirica* GroEl, OmpA (AA 1256–1734), and OmpB (AA 1210–1654) proteins (all from CRIE, Moscow, Russia) [23,24].

#### 2.5.2. Microarray Production and Processing

Antigens in predetermined concentrations (from 35 to 200 µg/mL) and human IgM control (Jackson ImmunoResearch, Chester County, PA, USA) solutions (in concentrations of 5, 10, and 50 µg/mL) were spotted, in triplicate, on the surface of aldehyde-activated VALS glass slides (CEL Associates, Los Angeles, CA, USA) using a sciFLEXARRAYER SX (Scienion AG, Berlin, Germany) to produce an ELISA-like multiplex assay in a microarray format. PBS (1x) was used as a printing buffer and negative control, while bovine serum albumin (Sigma-Aldrich Inc., USA) labeled with Cy3 NHS ester (Sigma-Aldrich Inc., USA) was used as an internal positive control (array border marker, ABM). After printing and overnight incubation in a humid chamber, slides were blocked for 1 h at 37 °C with a 0.5% BSA (Sigma-Aldrich Inc., USA) solution in PBS. Slides were stored at −20 °C. Immediately before use, microarrays were washed with PBST (1x PBS containing 0.01% of Tween 20) for 2 min at 37 °C with shaking at 500 rpm. Sixteen-well Fast Frames and Fast Slide Holders (Sigma-Aldrich Inc., USA) were used for well-formation. Serum samples (diluted 1:10 in PBS solution with 2% BSA) were added to each array and incubated for 30 min (37 °C with 500 rpm shaking). Next, all liquids were aspirated, and the wells were washed with PBST for 2 min (37 °C, 500 rpm shaking), followed by removal of all wash solutions (by aspiration) from all wells. After those wash steps, Cy3-conjugated goat anti-human IgM antibodies (50 ng/mL) (Jackson ImmunoResearch, USA), diluted in assay buffer, were added to each well and incubated for 30 min at 37 °C. After aspiration of liquids, a washing step was performed as described above. Slides were then removed from their frames and holders, washed with ultra-pure water (Milli-Q), and dried. The resultant fluorescent signals were measured on a MArS laser microarray scanner (Ditabis, Pforzheim, Germany).

#### 2.5.3. Data Quantification

Images were quantified using SpotScout software (Ditabis, Pforzheim, Germany) in accordance with its user manual. The obtained raw numeric data were processed as follows: human IgM dose-response calibration curves were fitted using a 3-parameter curve-fit for each array, and concentrations of IgM specific to recombinant antigens were interpolated from the human IgM calibration curves using ImStar software (CRIE, Moscow, Russia) for each array. Immunoglobulin M levels were calculated as micrograms per unit volume (µg/mL). Specific IgM levels to each antigen were considered significant if they exceeded 5 µg/mL [25,26,27]. This value was used as the permanent cut off for all antigens used in the microarray. Unification of the cut off value was achieved by varying antigen concentrations and sorption conditions during microarray development.

#### 2.5.4. Interpretation of Microarray Data

For all pathogens, a determination was made as to what criterion defined a positive result (Table 2).

Samples were considered positive for the presence of anti-*Borrelia* spp. IgM if antibodies were found against (1) at least one OspC antigen, in the absence or presence of antibodies against any other *Borrelia* spp. antigen; (2) at least two antigens belonging to different antigen groups (p41, p17, VlsE); or (3) against GlpQ antigen, in the presence of Abs against at least one antigen in a set (p39, p41, VlsE, Vsp1, Vlp5, Vlp15/16, Vlp18). All other samples were considered negative with respect to the presence of anti-*Borrelia* IgM.

Samples were considered positive for the presence of anti-*SFG* rickettsia antibodies if Abs were found against OmpA or OmpB antigens in the absence or presence of antibodies to GroEl. All other samples were considered negative with respect to the presence of anti-*SFG rickettsia* IgM.

Samples were considered positive for the presence of anti-*DENV* IgM if Abs were found against at least against one DENV NS1 antigen in the absence or presence of antibodies against E protein. Samples were considered positive for the presence of anti-*ZIKV* IgM or anti-*WNFV* IgM if Abs were found against ZIKV NS1 or WNFV NS1 antigen, respectively, in the absence or presence of antibodies against E protein.

Samples were considered positive for the presence of IgM against unspecified flaviviruses if antibodies were found against at least one DENV or ZIKV E antigen; samples were considered negative for the presence of IgM against flaviviruses if no antibodies against any of the antigens (DENV or ZIKV) were found.

Samples were considered positive for the presence of anti-*CHIKV* IgM if antibodies were found against the E1 or E2 antigens. In all other cases, samples were considered anti-*CHIKV* IgM negative.

Samples were considered positive for the presence of anti-*CCHFV* IgM if antibodies were found against (1) the NP and/or NPsh antigens in the absence or presence of antibodies against any of the other antigens or (2) against any number of G-antigens in the presence of antibodies to L-protein. In all other cases, samples were considered anti-*CCHFV* IgM negative.

Samples were considered positive for the presence of anti-*RVFV* IgM if antibodies were found against one of the NP or NPsh antigens in the absence or presence of antibodies against G2 antigen. All of the other samples were considered anti-*RVFV* IgM negative. For ZEBOV and MARV, samples were considered positive for the presence of anti-virus IgM if antibodies were found against their corresponding NP antigens.

Some previous evaluation work has been performed with the microarray [22,23,28,29,30,31,32]. Negative control samples were taken from Russian healthy donors. Presumptively positive samples were taken from our collection of PCR-confirmed cases of diseases either endemic in Russia (borrelioses, ricketttsioses, CCHFD, WNFD), imported to Russia (DFD, ZIKFD, CHIKD), or from Guinean patients (EVD). Positive serum samples were not available for RVFV and MARV. The array sensitivity and specificity are shown in Table 3. No cross-reactivity was observed between serum samples of patients with diseases caused by *Borrelia* spp., SFG *rickettsia*, CHIKV, CCHFV, or ZEBOV in relation to off-target pathogen antigens immobilized on the array. Due to the well-known phenomenon of high cross-reactivity between antibodies against different flaviviruses, interpretation criteria were adapted to minimize such effects. Only antibodies against their corresponding NS1 proteins were considered to be a marker of DENV or ZIKV infection (Table 2). Regarding ZIKV cross-reactivity, one anti-DENV IgM^+^ sample (of 33) and three anti-WNV IgM^+^ samples (of 48) cross-reacted with ZIKV proteins.

### 2.6. Statistical Analysis

The 95% confidence interval for a proportion was calculated according to refinements made by R. Newcombe on the procedure outlined by E. Wilson [33,34], using the calculator at https://epitools.ausvet.com.au/ciproportion (accessed on 16 April 2021). The significance of the difference between nominal variables (proportion of positive samples in different groups and subgroups) was estimated using Fisher’s exact test. The significance of differences between numeric scale variables (age of patients and interval between the date of disease onset and the date of sampling in different groups and subgroups) was estimated using nonparametric Mann–Whitney test and exact test (2-tailed) in IBM SPSS Statistics 19. The magnitude of differences between groups (effect size) was estimated for scale variables (Cohen’s *d*, Glass’s *Δ*) and nominal variables (odds ratio, Phi, Cramer’s V) [35].

## 3. Results

In total, 164 serum samples from 25 Guinean prefectures were analyzed (Table 4). Patient ages ranged from 2 to 75 years (median age 19 years, interquartile range 9–21). Women accounted for 44.5% ± 0.5% of the specimens and men for 55.5% ± 0.5%.

Using molecular methods (PCR), seven Lassa^+^ samples (4.3% ± 1.5% of the samples studied) were identified. All Lassa^+^ samples were confirmed using the ReLASV^®^ Pan-Lassa IgG/IgM ELISA Test based on GP-linked protein (Zalgen Labs LLC, Germantown, MD, USA). No other pathogens were detected during the PCR-based study of the samples. Using serological methods, detectable levels of specific serum IgM to antigens of one pathogen were observed in 45 of the studied samples (*n* = 157, excluding the 7 Lassa^+^ samples), as follows: IgM against YFV was found in 20 patients (12.2% ± 0.9% of all samples); IgM against DENV in 1 patient (0.6% ± 0.5% of all samples); IgM against ZIKV in 2 patients (1.2% ± 0.4% of all samples); IgM against CCHFV in 1 patient (0.6% ± 0.5% of all samples); IgM against SFG *rickettsia* in 8 patients (4.9% ± 1.2% of all samples); and IgM against *Borrelia* spp. in 13 patients (7.9% ± 1.3% of all samples). IgM against ZEBOV or MARV was not found.

In addition, we identified numerous samples with IgM against two or three pathogens together, as follows: IgM against SFG *rickettsia* and relapsing fever *Borrelia* spp. in 9 patients (5.5% ± 1.7% of all samples); IgM against DENV and CHIKV in 2 patients (1.2% ± 0.4% of all samples); IgM against *Borrelia* spp. and RVFV in 2 patients (1.2% ± 0.4% of all samples); and lastly, IgM against WNFV, SFG *rickettsia,* and relapsing fever *Borrelia* spp. in 2 patients (1.2% ± 0.4% of all samples). A total of 97 samples (59.1% ± 0.8% of all samples) were found to be negative using both molecular and serological methods (Figure 2).

The proportion of positive findings of laboratory diagnostics and the type of detected pathogens did not depend significantly on the place and time of sampling, gender, and age of patients (*p* > 0.1 in all comparisons). Even though differences between groups were present, their magnitude (effect size) should be considered “small” [35].

## 4. Discussion

All 164 clinical samples were obtained from patients unvaccinated against YFV with acute severe fever as their main clinical sign. Severe acute fever itself is not a unique, hallmark indication (pathognomonic sign) of any one illness or disease. As such, patient samples representing any number of viral or bacterial pathogens, including the Ebola, may have been present. Diagnosis was achieved in slightly over 40% of the studied samples; viral and bacterial pathogens were both identified. Similar data was obtained during an investigation of causative agents of acute fever in samples collected in the neighboring West African country of Mali: evidence of viral or bacterial infection was found in 39.9% of samples (14.4% *Leptospira* spp., 7.7% DENV, 5.3% CHIKV, 0.27% WNFV, 7.2% hantaviruses, 0.27% LASV, and 4.8% CCHFV IgM-positive, respectively) [36].

Among viral pathogens, most were attributable to YFD (12.2%) and LFD (4.3%), which is not surprising given that both are endemic to the territories of Guinea and neighboring Sierra Leone [5,37,38]. In 2008, Guinea reported about six confirmed cases of YFD (two in each of the Faranah, N’zérékoré, and Kankan health districts) and about 41 suspected YFD cases (21, 14, and 6 in Faranah, N’zérékoré, and Kankan health districts, respectively), four of whom died due to fever and jaundice [39]. YFD and LFD cases were not linked epidemiologically. Natural foci of YFV and LASFV exist in Guinea, and these contribute to a sporadic background incidence. This background makes outbreaks and epidemics of these viruses (YFD and LFD) continuously possible. We considered the LFD^+^ and YFD^+^ case determinations to be reliable based on methodology. Lassa^+^ samples were confirmed using both PCR and ELISA, while YFD^+^ samples were diagnosed using the WHO-recommended ELISA method.

Furthermore, CCHFD^+^ (0.6%), ZIKFD^+^ (1.2%), and DFD^+^ (0.6%) cases were identified. Therefore, these viral infections were demonstrably present among humans in Guinea. This fact should be taken into account during differential diagnosis of acute febrile illnesses. Ideally, field results would be further confirmed with high confidence using a plaque reduction neutralization test (PRNT). It is hardly feasible, however, in the low-resource settings in which studies are being carried out in Guinea. Notably, the first human WND case in Korea was imported from Guinea and confirmed by PRNT in a Korean laboratory [40]. In addition, there was evidence of ZIKV circulation in Senegalese vectors and the population, suggesting the possibility of such circulation in Guinea due to similarity in climate and habitat type [41]. The seroprevalence of DENV, CHIKV, and ZIKV in surrounding West African countries was found to be 10–30%, 30–40%, and 3–5%, respectively (the possible cross-reactivity of detected IgG antibodies within flaviviruses and alphaviruses was not considered) [42].

Obviously, all of these infections may be imported from Africa. In Europe, the most affected are southern countries where importation may lead to autochthonous outbreaks [14]: Spain, France, Croatia, Greece, and Italy, in particular. From 2008–2011, 109 imported cases of DENV infection and 21 imported cases of CHIKV infection were reported to the Italian National Institute of Health [15]. When the *National Plan on Human Surveillance of Vector-borne Diseases* was implemented, the Italian National Reference Laboratory for Arboviruses diagnosed 68 laboratory-confirmed imported cases of DENV infection, 35 imported cases of CHIKV infection, along with the detection of the first four confirmed ZIKV cases, in the period from July 2014 to October 2015 [16]. The number of DVD cases in Russia is even higher. Since its first detection in the country, more than 1500 clinical DVD cases have been officially registered (2012–2019), with a maximum of 415 cases in 2019; all of them were imported [43,44,45]. According to estimates by Napoli et al. [15], the number of DENV-exposed travelers may be about 20-fold higher.

Although most imported cases of vector-borne infection come from popular recreational areas in Thailand, Maldives, Vietnam, etc., in absolute numbers [15,16,43,44,45], the relative risk of travelers contracting zoonoses may be higher in West Africa. 

In addition to illnesses of viral etiology, cases of SFG rickettsiosis (4.9%) and relapsing fever caused by *Borrelia* spp. (7.9%) were identified. Mixed infections, mainly *Borrelia* spp. with SFG rickettsiosis (5.5%), were seen. Other mixed infections were also seen: DENV and CHIKV; RVFV and *Borrelia* spp.; and three together (WNV, SFG *rickettsia,* and relapsing fever *Borrelia* spp.). These findings suggest that Guineans are likely attacked by both mosquitoes and ticks, and that these attacks may be occurring simultaneously or over short time frames. In total, 15.9% of acute febrile illnesses had bacterial etiology in our study. Similarly, DNA from at least one pathogenic bacterium were identified in 80/440 (18.2%) of the samples from febrile patients in Senegal (35, 30, 23, 2, and 1 cases for *Borellia crocidurae, Rickettsia felis*, *Bartonella* spp., *Coxiella. burnetii,* and *Tropheryma whipplei* identification, respectively [46].

Bacterial infections can also be imported. Evidence of *B. crocidurae* infection has been noted in travelers returning to France and Italy from Senegal and Mali [47,48,49]. Spotted fever group rickettsioses, first of all, *R. africae* infection, have been diagnosed in travelers returning to the U.S. from Liberia, Gambia, and other African countries [50]. *R. typhi* infection was found in a traveler returning to Spain from Senegal [51].

The geographic distribution of positive samples was uneven (Table 4). Most positive samples (all types) were obtained from lower Guinea (29 positive samples of 85, or 37%), while 14 positive samples were obtained from middle Guinea (44% positive of 32 samples) and upper Guinea (61% positive of 23 samples) each. In addition, seven positive samples (29% positive of 24 samples) were obtained from forested Guinea. Although noticeable, differences in the proportion of positive samples in different geographic areas did not reach statistical significance (*p* = 0.12, Fisher’s exact test), and effect size was small (Cramer’s V = 0.19). For some pairwise comparisons, odds ratios were rather high: 3.8 (*p* = 0.04) and 2.7 (*p* = 0.06) when comparing upper Guinea versus forested Guinea or lower Guinea, respectively.

No significant socio-demographic differences among positive samples were seen.

In a sizable number of cases (approximately 60%), a diagnosis was not determined. Several factors may have been contributing to this. One potential reason may have been sample handling issues, such as sub-optimal sample collection time frames, improper storage conditions, or deficiencies related to delivery of materials to the laboratory. It is also possible that the set of infectious agents (of various etiologies) causing fever was broader than the pathogen panel used to analyze patient samples. For example, *Leptospira* spp., *Bartonella* spp., hantaviruses, as well as Bombali virus can also cause febrile illness [36,46,52]. Perhaps it is appropriate here to quote verbatim the results of a long-term, large-scale study of the Soviet era [53]: 

“In 1978–1991, the USSR–Guinea Virological and Microbiological Laboratory functioned in Kindia, the Republic of Guinea. … About 74,000 mosquitoes, 100,000 Ixodidae ticks, 1500 wild birds, 2700 bats, 106 monkeys, 308 other mammals, and 927 blood samples collected from febrile patients were examined in 1978–1989, using inoculation of new-born white mice. As a result of this work, 127 strains of the following arboviruses were isolated: Chikungunia (one strain), Dengue 2 (four), Saboya (seven), Wesselsbron (one), Bunyamwera (four), M’Poko (five), Rift Valley Fever (six), CHF-Congo (nine), Dugbe (22), Bhanja (six), Forecariah (two), Jos (26), Abadina (15), Kindia (two), Ark 6956 (one), Fomede (two), Bluetongue (nine), Mossuril (two), AnK 6009 (one), and Kolente (two). Dengue 2, Wesselsbron, Bunyamwera, M’Poko, Kindia, and Mossuril viruses were isolated from mosquitoes. Ixodidae ticks were sources for isolation of Chikungunia, Saboya, CCHF, Dugbe, Bhanja, Forecaciah, Jos, Abadina, Kindia, Ark 6956, Fomede, Bluetongue, and Kolente viruses. Saboya, RVF, Fomede, Kolente, and AnK 6909 were isolated from bats (Chiroptera); Saboya, Abadina, and Bluetongue viruses were isolated from birds. One strain of Dugbe virus was originated from the brain of Cercopithecus patas. Bunyamwera and Abadina viruses were isolated from the blood of two febrile patients. Serological identification of many strains was kindly conducted at the Pasteur Institute, Dakar (J. P.Digoutte) and some at the YARU, USA (R. Shope)”.

Not all of these virus species are pathogenic to humans, of course, but some of them might potentially be responsible for undiagnosed febrile illness cases in our study or other studies [5,38]. T. Pierson and M. Diamond have considered African Wesselsbron arbovirus and Zika-like Spondweni virus to have potential as newly emerging flaviviruses [54].

Another problem is that the currently available nucleotide sequence data on African pathogenic strains are scarce. For example, of the 125 DENV and CHIKV “reference” nucleotide sequences used to clarify the origin of DENV or CHIKV causing human cases imported into Italy [16], only eight “reference” isolates originated from Africa. Therefore, unidentified target-proximal genetic variability can hinder PCR diagnostics, leading to negative results, even for diseases caused by known pathogens. Possibly a Pan-Degenerate Amplification and Adaptation (PANDAA) approach [55] could be useful in this situation.

The main limitation of our study was its modest statistical power; we were not authorized to collect enough clinical samples by ourselves and used additional samples donated by the collection of the Hemorrhagic Fevers Research Project in Guinea. As a result, some additional pathogens might not have been found (by random chance), or conversely, the frequency of other pathogens may have been overestimated. This study did, however, reveal a number of possible febrile illness agents and their relative importance in Guinea. As such, it outlined the way for further research. Such assessments can identify regions where needs and provisions do not align. These areas should be targeted for future strengthening and support of public health, as has been encouraged by a huge team of experts in a multistage analysis [56].

Therefore, determining exactly which infectious pathogens are most relevant to Guinea is an extremely important step in terms of improving the local health care system and facilitating differential diagnosis of acute fevers. Clarifying data, from this work and future research, will be useful in the event of new Ebola Virus Disease outbreaks. Knowledge of the spectrum of zoonoses endemic in Guinea is also necessary to improve the quality of differential laboratory diagnostics for rapid identification of imported cases among people who have arrived from West Africa to Europe. In addition, such knowledge could be necessary, both for assessing the risks of people traveling to Guinea and for planning preventive measures, such as vaccination.

## 5. Conclusions

We characterized, in Guinea, a number of infectious diseases that present with severe fever, including YFD, LFD, CCHFD, DENFD, SFG rickettsiosis, and relapsing fevers caused by *Borrelia* spp. In addition, several co-infections were identified by microarray (DENV-CHIKV, RVFV-*Borrelia* spp., and WNFV-SFG *rickettsia-Borrelia* spp.). Although CHIKV and RVFV were only identified in co-infections (CHIKV+other or RVFV+other), it was clear that both etiologic agents could exist separately. The immobilized protein microarray presented here was determined to have quite acceptable sensitivity and specificity, and successful express identification of several highly dangerous pathogens was demonstrated. Still, the methods used did not identify a pathogen in a number of severe fever cases. Therefore, further work is needed to establish a list of the most important etiologic agents relevant to Guinea to improve existing methods and to develop new diagnostic tools.

## Figures and Tables

**Figure 1 ijerph-18-06022-f001:**
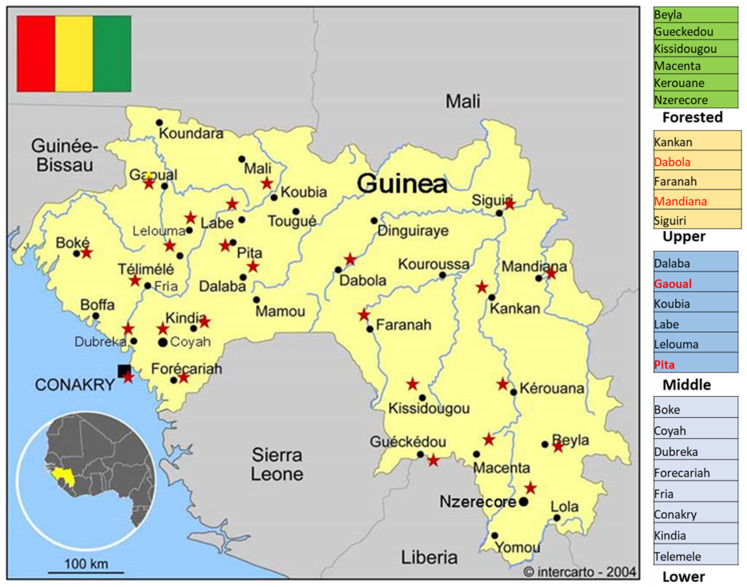
Map of Guinea. Districts where clinical samples were collected are marked with red stars.

**Figure 2 ijerph-18-06022-f002:**
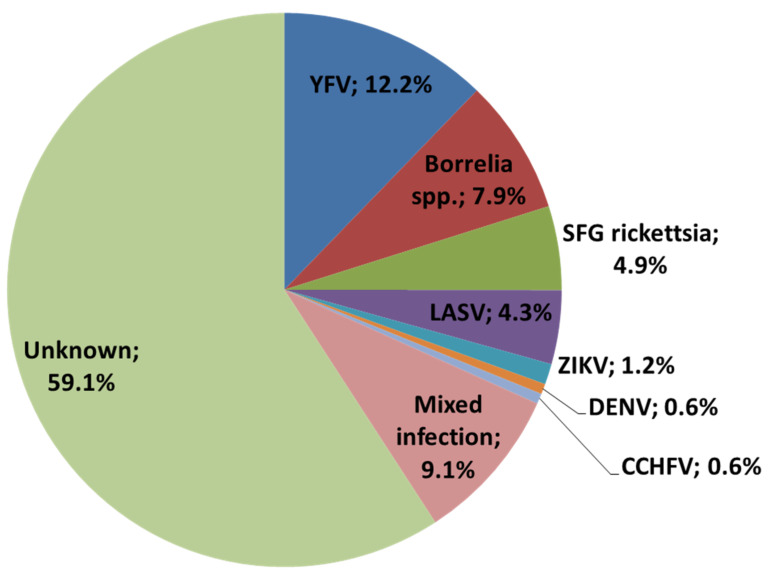
Distribution of zoonotic diseases in the study group, in percentages.

**Table 1 ijerph-18-06022-t001:** Methods used to diagnose infectious diseases.

Infections and Methods *	YFD	CCHFD	LFD	DFD	ZIKFD	EVD	MFD	WNFD	RVFD	CHIKF	SFG Rick.	*Bor.* spp.
Polymerase chain reaction		PCR Amplisens	PCR Ölschläger	PCR Amplisens		PCR Amplisens	PCR Amplisens	PCR Amplisens				
ELISA	MAC-ELISA IgM capture assay		ReLASV^®^ Pan-Lassa IgG/IgM ELISA									
Protein microarray, IgM		Array		Array	Array	Array	Array	Array	Array	Array	Array	Array

* Explanation of abbreviations and details of the methods used are given in the text (Section 2.3, Section 2.4 and Section 2.5). YFD (yellow fever disease), ZIKFD (Zika fever disease), RVFD (Rift Valley fever disease), CHIKF (Chikungunya fever), SFG rick. (spotted fever group rickettsias), *Bor.* spp. (relapsing fever *Borrelia* spp.).

**Table 2 ijerph-18-06022-t002:** Algorithms for the interpretation of microarray data.

Antigens Used in Microarray	IgM Antibody to:	IgM Antibody to:	IgM Antibody to:	Conclusion *
*Bor.* spp. antigens	OspC alone or with any other antigen	two of three (p41, p17, VlsE)	GlpQ and at least one antigen in a set (p39, p41, VlsE, Vsp1, Vlp5, Vlp15/16, Vlp18)	any of the options on the left: *Bor.* spp. IgM present; none of the options on the left: *Bor.* spp. IgM absent
SFG rickettsia antigens	OmpA	OmpB		any of the options on the left: SFG rickettsia IgM present; none of the options on the left: SFG rickettsia IgM absent
ZEBOV antigen	NP			if present—ZEBOV IgM present; if absent—ZEBOV IgM absent
MARV antigen	NP			if present—MARV IgM present; if absent—MARV IgM absent
CCHFV antigens	NP and/or NPsh	any number of G-antigens and L-protein		any of the options on the left: CCHFV IgM present; none of the options on the left: CCHFV IgM absent
WNFV antigen	NS1			if present—WNFV IgM present; if absent—WNFV IgM absent
DENV antigens	any NS1			if present DENV IgM present; if absent—DENV IgM absent
ZIKV antigen	NS1			if present—ZIKV IgM present; if absent—ZIKV IgM absent
unspecified flaviviruses	any DENV E	ZIKV E		any of the options on the left: IgM to unspecified flaviviruses present
CHIKV antigens	E1	E2		any of the options on the left: CHIKV IgM present; none of the options on the left: CHIKV IgM absent
RVFV antigens	NP	NPsh		any of the options on the left: RVFV IgM present; none of the options on the left: RVFV IgM absent

* Explanation of abbreviations and details of the method used are given in the text (Section 2.5).

**Table 3 ijerph-18-06022-t003:** Sensitivity and specificity of the protein microarray assay.

Pathogen	Number of Samples from PCR-Confirmed Patients	Sensitivity, %, and 95% Confidence Interval (In Parentheses)	Number of Samples from Healthy Donors	Specificity, %, and 95% Confidence Interval (In Parentheses)
*Borrelia* spp.(acute)	132	66 (57.5–73.4)	300	97 (94.4–98.4)
SFG rickettsia	100	72 (62.5–80.0)	200	98 (95.0–99.2)
DENV	60	72 (59.2–81.5)	100	98 (93.0–99.5)
ZIKV	30	83 (66.4–92.7)	100	98 (93.0–99.5)
CHIKV	4	75 (30.1–95.4)	100	98 (93.0–99.5)
CCHFV	20	85 (64.0–94.8)	100	98 (93.0–99.5)
WNFV	12	67 (39.1–86.2)	100	98 (93.0–99.5)
RVFV	not available	-	100	98 (93.0–99.5)
ZEBOV	3	100 (43.8–100)	100	98 (93.0–99.5)
MARV	not available	-	100	98 (93.0–99.5)

**Table 4 ijerph-18-06022-t004:** Analysis results from acute febrile illness patient samples.

Part of Guinea	Total Number of Samples	Males	Females	YFD	CCHFD	LFD	DFD	ZIKFD	SFG Rick.	*Bor.* spp.	*Bor.* spp. + SFG Rick.	DFD + CHIKFD	RVFD + *Bor.* spp.	WNFD+ *Bor.* spp. + SFG Rick.
Forested	24	15	9	1	0	0	0	0	3	3	1	0	0	0
Upper	23	11	12	1	0	4	0	1	2	1	3	0	0	2
Middle	32	17	15	5	1	2	1	0	0	1	1	1	2	0
Lower	85	48	37	13	0	1	0	1	3	8	4	1	0	0
Total	**164**	**91**	**73**	**20**	**1**	**7**	**1**	**2**	**8**	**13**	**9**	**2**	**2**	**2**

## Data Availability

Data collection and data handling procedures strictly complied with the EU’s General Data Protection Regulation (GDPR) and the Helsinki Declaration.

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
