# Peer review of "Differential Laboratory Diagnosis of Acute Fever in Guinea: Preparedness for the Threat of Hemorrhagic Fevers"

_ijerph, 2021, doi:10.3390/ijerph18116022_

Round 1

Reviewer 1 Report

This paper provides a much needed documentation on fevers of unknown origin and their wide-spread prevalence. However, this paper requires significant revision in order to more clearly present the results and discussion. 

Introduction

-Paragraphs 2-4 basically state the same information in different ways and should be better consolidated. 

  • Overall comment - the role of Ebola in this paper, it is continually brought up and included for testing, but it does not look like there were any positive case? But the data wasnt presented. Also, none of the enrolled participants were enrolled due to suspected Ebola. Paragraph 7 makes it sound like participants may have been suspected Ebola cases, but were actually selected from suspected yellow fever cases. The general title indicates also Ebola, but then no results from the Ebola testing are actually presented in the results. The title seems a bit misleading.

Methods

-Table 1 is actually results and should be moved to that section. I think the centers could be collapsed by region - unless there are differences between the sites that should be noted, Maybe include the number of samples collected in each site as a new table 1? 

-Paragraph 2 - distribution of women should be a part of the results section

-EVD and MARV is mentioned as two of the diseases tested for (either both by PCR and or microchip), but there are no results presented for this or no further mention of why the results were not included. Also for these it looks like the NP as opposed to the GP protein were the point of interest, what was the reason for this?

-Also how did the results of the PCR and microchip compare and why are they not better labeled in results? Are the results combined from both methods? It looks like the results in table 1 are from PCR? Except for the bacterial diseases? 

-Section 2.5.4 describes how cut-offs were defined, but there are no actual values listed, what constitutes "antibodies being found" - this is not clear and could it present issues with areas where there could be a higher background? The authors should specifically list the cut off used for each test. Also antibodies from which test? Both tests? This needs to be clarified.

2.6 - is this just for the sens/spec results? This should be stated as it does not look like there are any other statistical analysis completed other than just basic frequencies, or is this what is in ref to the % +/- in the results, - what program was used for this?

Results

  • Should start this off with the overall samples collected/analyzed and the male/female distribution (listed in methods). Also this should correspond to the table 1 results, here you present % and in the 1st table is is just numbers. These could be collapsed into region and have an overall total. This will help the reader better visualize. It will also be equivalent to figure 2.
  • An additional table could be the demographic table of the participants, what else is known about them? Location, gender, age? reason admitted? Ect. This could also prove helpful for understanding undiagnosed cases, is there more in 1 region than another more male/female, ect. 

Discussion

  • 1st sentence of discussion just repeats what was in the methods section.
  • -Authors state: "As such, patient samples representing any number of viral or bacterial pathogens, including the Ebola, may have been present." - but they do not show any results indicating people had probable ebola infection, and actually do not show any results related to ebola. I would remove this statement. 
  • The authors state: "especially in rural areas." But they did not provide any results related to this statement on differences between rural/urban areas
  • The authors state: "The geographic distribution of positive samples was uneven. Most positive samples (all types) were obtained from Lower Guinea (29 positive samples),while 14 positive samples were obtained from Middle Guinea and Upper Guinea each. In addition, 7 positive samples were obtained from Forested Guinea. N" However, they do not give indication in this statement if there were different proportions collected, was the most positive samples from where they collected the most samples? Proportionally how does this look? I could do the calculation on my own, but this burden should not lie on the reader, but instead the author to provide clear results and discussion that is related to data presented in the results. And the next sentence does present this, but it conflicts with the statement before, as a different areas actually has higher % of positive cases, even though they do not have the most cases. 

Reviewer 2 Report

1) Introduction, last paragraph: the abbreviation 'EVD' is used here for the first time, so it should be explained here.

2) Table 1: Abbreviations (YFD, CCHFD, etc) should be explained as footnotes to the Table.

3) Reference List: The citation reference 16 is not complete. 

4) Section 2.5.4: The information in this section could be presented in a table, rather than in the main text. The authors should consider this.

5) The abbreviations EBOV and MARV should be explained at first use.

6) Results, first paragraph: the authors should include a reference to Table 1.

7) Table 2: The authors should clarify the use of the character ' ± ' and the sign for division. Are these the correct characters here?

8) Informed consent: The authors state that informed consent was not required, but in the Methods the authors mention the sample collection. The authors should clarify whether the samples were collected for this study or another study, and relevant ethics information.
